# Proteomic and Genomic Studies of Micronutrient Deficiency and Toxicity in Plants

**DOI:** 10.3390/plants11182424

**Published:** 2022-09-16

**Authors:** Suchismita Prusty, Ranjan Kumar Sahoo, Subhendu Nayak, Sowmya Poosapati, Durga Madhab Swain

**Affiliations:** 1Department of Biotechnology, Centurion University of Technology and Management, Bhubaneswar 752050, Odisha, India; 2Division of Health Sciences, The Clorox Company, 210W Pettigrew Street, Durham, NC 27701, USA; 3Division of Biological Sciences, Cell and Developmental Biology Section, University of California, San Diego, CA 92093, USA

**Keywords:** micronutrients, proteomics, genomics, nutrient toxicity, nutrient deficiency

## Abstract

Micronutrients are essential for plants. Their growth, productivity and reproduction are directly influenced by the supply of micronutrients. Currently, there are eight trace elements considered to be essential for higher plants: Fe, Zn, Mn, Cu, Ni, B, Mo, and Cl. Possibly, other essential elements could be discovered because of recent advances in nutrient solution culture techniques and in the commercial availability of highly sensitive analytical instrumentation for elemental analysis. Much remains to be learned about the physiology of micronutrient absorption, translocation and deposition in plants, and about the functions they perform in plant growth and development. With the recent advancements in the proteomic and molecular biology tools, researchers have attempted to explore and address some of these questions. In this review, we summarize the current knowledge of micronutrients in plants and the proteomic/genomic approaches used to study plant nutrient deficiency and toxicity.

## 1. Introduction

Essential nutrients (macro and micro) are required by plants for appropriate functioning and development. The significance of micronutrients in plant nutrition is well recognized. Micronutrients comprise less than 1% of the dry weight of most plants and are vital for their growth [1]. Plant classified micronutrients include boron, chlorine, copper, iron, manganese, molybdenum, nickel and zinc, which are vital for completion of the plant’s life cycle (Figure 1) [2]. They are also essential to maintain the stability of proteins and cellular structures. Through their interactions with other physiologically active molecules and enzymes, micronutrients play an essential role in the biosynthesis of proteins, nucleic acids, cofactors, carbohydrate metabolism, lipid metabolism, stress tolerance, chlorophyll maintenance, electron transport, anti-oxidative systems and much more. Hence access to micronutrients is critical for optimum crop nutrition and development (Figure 2) [3]. The bioavailability of micronutrients is heavily influenced by climatic factors like drought, severe rain, waterlogging or salinity [4]. Energy metabolism, primary and secondary metabolism, cell defense, gene expressions and regulations, hormone sensing, signal transduction, and reproduction are all influenced by micronutrients. The proportion of micronutrients in soil is determined by the geological substrate and pedogenesis management strategies. The ideal concentration of each micronutrient in the crop is influenced by chemical and physical features of the soil, such as soil pH, nutrient availability, clay minerals, microbial activity, amount of organic matter in the soil, quantity of other nutrients, and other factors that might affect micronutrient absorption and efficacy (Table 1). Although the fraction of micronutrients available in the soil may exceed a single plant’s requirements, the accessible proportion may be insufficient for all, resulting in nutritional deficiency in the crop. In Indian soils, improper nutrient management leads to multi-nutrient deficits [5]. Growth of the plant is constrained either by lack of micronutrients, nutrient toxicity or soil conditions.

Micronutrient depletion develops in the soil as a result of farming techniques, such as intensive farming, monocultures, and acid soil liming, which affect the plants in various ways (Table 2, Figure 3). When nutrient demand exceeds the rate of supply, the plant frequently switches to alternative metabolic pathways, which are often dependent on limiting micronutrients. A decrease in one micronutrient content might reduce the bioavailability of other nutrients in the soil [6]. Although, in certain cases, plants seem unable to detect a deficit because micronutrient availability is impacted by organic matter content, soil pH, adsorptive surfaces, and other biological, chemical, and physical environmental conditions. Prolonged negligence of micronutrient supplementation and aversion of organic fertilizers are major contributors to micronutrient insufficiency and plants respond to these conditions in a variety of ways, by reprogramming transcriptional and translational modifications [2]. Identifying metal-tolerant genes and/or proteins is the first step in deciphering the pathways linked with micronutrient stress tolerance [7].

Micronutrient toxicity also leads to various phenotypic, as well as genotypic, changes in the plants (Table 3). When the internal quantity of the micronutrients surpasses the threshold, they cause phytotoxicity. To deal with the scenario, a plant develops a set of tactics at both the proteomic and genetic levels (Table 4 and Table 5). Plants with tolerance to heavy metals have also been examined for proteomic alterations, revealing one more field of molecular application and its regulation [60]. Some plants respond to nutrient toxicity by increasing nutrient efflux and activating detoxifying pathways, whereas others can tolerate high concentrations of certain micronutrients because they have developed systems to store these and utilize them for defensive functions (Figure 4).

Balancing nutritional stressors leads to a multi-genetic response that causes several changes in proteins and genes, which has a direct impact on nearly all biological activities in a live cell. Proteomic and genomic methods can, thus, be instrumental in identifying molecular responses to nutritional stress [2]. In this review, we summarize the representative roles of different micronutrients on plants. We also provide the current information available from the proteomic and genomic studies of nutritional stress along with explaining how current research is addressing the issue of mineral toxicity in agricultural soil.

## 2. Micronutrients and Their Roles

### 2.1. Boron

Boron plays a role in cell walls by cross-linking pectin polysaccharides. About 26 boron-binding proteins have been found in Arabidopsis, and 9 have been reported in maize [81]. They are involved in stress responses, DNA processing, cell cycle, ions and cell transport, detoxification, antioxidation, energy and amino acid metabolism, coenzyme biosynthesis, nucleotide metabolism, and cytoskeleton [82].

### 2.2. Chlorine

Chlorine is essential for photosynthesis because of its involvement in the opening and closing of the stomata. Around 130 organic compounds in plants contain chlorine [83]. Chlorine influences germination and energy transfer by activating key enzymes involved in starch metabolism.

### 2.3. Copper

Copper is involved in a variety of pathways, including respiratory electron transport chains, ethylene sensing, cell wall metabolism, oxidative stress protection, molybdenum cofactor biogenesis, cell wall composition, and lignin synthesis. Cu homeostasis not only ensures adequate copper levels, but also protects against the detrimental consequencesof excess copper, which causes the formation of reactive oxygen species (ROS) via the Fenton reaction Cu-binding proteins, which are active in anti-oxidative defense, protein modification, and metabolic control, and overexpressed in response to higher Cu concentrations [84].

### 2.4. Iron

Iron (Fe) is an essential and the most common transition metal in plants [85]. Iron-containing proteins are involved in DNA stability and repair, cellular respiration, photosynthesis, oxygen transport and intermediate metabolism in plants. As iron oxides have a limited solubility, the availability of iron decreases as oxygenation increases.

### 2.5. Manganese

Manganese (Mn) is readily absorbed by plants in acidic soils because it is more accessible. Manganese is essential for photosynthesis, respiration and activation of several enzymes. Natural resistance-associated macrophage protein (Nramp) transporters, cation diffusion facilitator (CDF) transporter family, P-type ATPases, cation/H+ antiporters, and ZIP transporters are among some of the protein families involved in the transport and absorption of Mn^2+^ [86].

### 2.6. Molybdenum

The availability of molybdenum is necessary for plant growth which is determined by a variety of parameters, such as soil pH, concentration of adsorbing oxides (e.g., Fe oxides), water drainage etc. Involvement of Molybdeno-enzymes is reported in the fixation, assimilation and reduction in nitrogen (by nitrate reductase, nitrogenase), catabolism of purine (by Xanthine oxidase/dehydrogenase), sulfur-metabolism (by sulfite oxidase) and synthesis of indole acetic acid and abscisic acid (by aldehyde oxidase). These have a role in stress responses, sulfur metabolism, nitrogen uptake and phytohormone production [87]. In order to having biological action, molybdenum must be linked to pterin, producing a molybdenum cofactor (Moco). Moco forms molybdeno-enzymes when it binds to molybdenum-requiring enzymes like nitrate reductase, sulfite oxidase, xanthine dehydrogenase, and aldehyde oxidase [88].

### 2.7. Zinc

Zinc is acquired from the soil by acidification and H+ extrusion by proton pumps in the plasma membrane, a process that is similar to iron absorption in strategy I plants. When a plant is deficient in zinc, it increases absorption from the root symplast [89]. Deficiency indicates a delay in maturation. The majority of enzymes involved in the control of DNA transcription, RNA processing, and translation require zinc for their activity.

## 3. Molecular Approaches for Understanding Micronutrient Stress Mechanisms in Plants

Micronutrient insufficiency and toxicity might be detected using both proteomic and genomic investigations [6]. The distribution of micronutrients all throughout the soil profile influences their availability to plants. Plants adapt dynamically to maintain nutrient supply and demand in the appropriate range. Proteomics is a popular molecular method for defining full proteomes at the organelle, cell, or tissue level. It is also helpful for comparing proteins under various adverse environmental conditions [90]. Sub-proteome analysis of nutrient deficit plants is also becoming popular, which includes analysis of apoplastic fluids, root plasma membrane, microsomal shoot fractions, phloem saps detergent-resistant membranes, thylakoid membranes and root hairs [91]. Proteins are the key molecules involved in several biochemical processes, and, as a result, a thorough understanding of stress induced genomic and proteomic changes aids in deeper understanding of the stress induced pathways. Changes in proteomic expressions are correlated with changes in the gene, transcriptome and metabolism levels. However, alterations at the transcriptional level may not always correspond to changes at the proteomic level. Protein expression is regulated not just at the transcriptional level, but also at the translational and post-translational stages, despite the perception of reciprocity between mRNA and protein [92]. As a result, information gathered at the translational and post-translational levels can provide more insight into protein responses, their modifications and functional relationships than genome-based predictions can provide. As proteins act as direct mediators of response, examining these alterations at the proteome level is critical [93]. These variations are reflected in the proteome compositions, hence proteomic investigations might be useful in identifying important protein components. These could be used as possible biomarkers in the underlying process. Current proteomic investigations have mostly focused on detecting quantitative changes and have relied on comparative proteomic techniques that include two-dimensional gel electrophoresis (2-DE) followed by mass spectrometry analysis (MS). This should lead to a better knowledge of the interactions between various elements, as well as the plant’s responses to environmental conditions at various phases of growth and development.

The genome of a life form is consistent, but the proteome is much more complex because protein expression changes with time and environmental factors [94]. Several researchers used transcriptome analysis to examine the expression patterns of genes in plants under heavy stress throughout the last decade. Gene expression at the mRNA level may be used to determine plant responses to a micronutrient buildup (Table 5) [95]. Furthermore, transcriptional analysis has some drawbacks, such as a lack of correlation between changes in mRNA expression and changes in their related proteins. By combining this method with genotyping technologies, researchers are able to quickly identify genes and networks that coordinate accumulation of elements in plants. A detailed summary of different genomic and proteomic studies carried out using different micronutrients is listed below.

### 3.1. Boron

Boron deficiency develops as a result of decreased root respiration, advanced cellular transport, rise in antioxidants and ROS-scavenging proteins [96]. Total proteins in boron deficit white lupin (*Lupinus albus*) root extracts were analyzed using 2D-PAGE, and 128 proteins were determined using mass spectrometry, all of which were involved in cell structure and metabolism, protein metabolism, energy pathways and defense mechanism [97]. ITRAQ study of the roots of *Citrus sinensis* seedlings subjected to Boron deficiency revealed a rise in level of 164 proteins, as well as reduction in level of 225 proteins [96]. Many of these proteins were involved in signal transduction, cell transport, stress responses, nucleic acid metabolism, protein metabolism, carbohydrate metabolism, biological regulation, lipid metabolism, cytoskeleton metabolism and energy metabolism.

Boron-toxicity-responsive proteins have been identified through MS analysis in research carried out on the leaves of *Citrus sinensis* and *Citrus grandis* (B- tolerant citrus species). Toxicity to boron is relatively common in alkaline and saline soils. Boron toxicity increases the amount of PSI type III CAB in barley leaves [98]. According to the 2D-PAGE research of boron toxicity-responsive proteome in oranges, it was hypothesized energy metabolism and photosynthesis might have resulted in increased CO_2_ absorption and, hence, resulted in a better continuation of energy balance [82]. Proteins associated to ROS-scavengers were highly accumulated in Boron-toxic *C. grandis* compared to *C. sinensis* to overcome oxidative stress, according to the analysis of root samples [99]. In a separate study, root systems of these citrus species were tested to study the toxicity of boron. In both species, there were 44 up- and 66 down-regulated genes, with Root Hair Defective 3 expressing in *C. sinensis* and villin4 being repressed. The discovery of boron-toxic-responsive genes involved in the lipid, nucleic acid and energy metabolisms helped researchers to further understand the mechanisms behind boron-toxicity in citrus species [100]. Using the SHB1/HY1 gene and increased levels of BOR4 expression, researchers established the boron homeostasis concept in Arabidopsis root samples [72]. According to the findings, excessive boron absorption in plants promotes excess transcription of the BOR4 gene, an efflux type boron transporter, encouraging the exclusion of excessive boron. In Oryza sativa, the genetic diversity linked with boron tolerance was investigated by looking at genes involved in biochemical binding, transport, transcriptional control, and redox homeostasis [73].

### 3.2. Chlorine

The majority of soils are not chlorine deficit except for sandy soils or soil covering heavy rainfall areas. Reduced leaf surface area, withering of the plant, and limited, highly branching root systems are some of the most common symptoms associated with chlorine deficiency. Near the leaf’s tip, little patches of pale green chlorotic tissue emerge between the major veins. The old leaves show the first signs of chlorine deficit (Figure 3) [2]. The total chlorine content in the plant is very low. Chlorine accumulation in plants occurs in specific tissues, notably leaves or single cells (for example guard cells), causing toxicity.

In plants, chlorine-anion transporters have been found as homologous genes. They include ATP binding cassette (ABC) transporters, chloride intracellular channel (CLIL) and nucleotide sensitive–chloride conductance regulator protein (ICln) [101]. As proposed by a few researchers with studies on rice (7 members, OsCLC1-7) and Arabidopsis (7 members, AtClCa-g), members of the vast Chloride Channel (CLC) family were found in several organs [102]. They have been reported to encode anion channels/ion transporters that are essential in nitrate homeostasis [103]. Proteins from the CLC family of Arabidopsis were found in a variety of membranes, including the chloroplast membranes (AtClCd and AtClCf) (AtClCe), vacuolar membranes (AtClCa) and Golgi vesicles (AtClCd and AtClCf), and guard cells (AtClCc). AtClCa expression patterns in roots are very significant. They function as a 2NO3 K/1HC exchanger, capable of accumulating nitrate in the vacuole using electrophysiological and genetic approaches in combination. The Cation-Cl-Cotransporters (CCC) proteins of *A. thaliana* (At CCC) catalyzed the coordinate symport of K^+^, Na^+^, and Cl^−^ in *Xenopuslaevis* oocytes [104]. Recent advances in the identification of novel transporters of Cl^−^ have been reviewed by Li et. al., 2017 [67].

### 3.3. Copper

The research found that a copper deficit increased the expression of Cu transporters in roots (copper transporter 2, COPT2) and leaves (heavy metal ATPase 1, HMA1). A 2D-PAGE proteomic study of Cu-deficient leaves identified roughly 33 proteins that were differentially regulated and approximately half of these were found in chloroplasts. About 11 differently expressed proteins needed Cu for protein synthesis or its function, and the Cu-dependent upstream or downstream enzymes were likewise differentially expressed. 2D-PAGE analysis on alfalfa stems indicated the accumulation of specific Cu chaperones. Cu deficiency led to an increase in the transcript levels of pyruvate ferredoxinoxido reductase (PFR1), several hydrogenases (HYD1, HYDEF, HYDG) and putative hybrid cluster protein (HCP2, HCP3), all of which are linked to anaerobic responses [105]. As the proteins associated with these transcripts cannot function in aerobic circumstances, due to their O_2_-labile active-site clusters, it resulted in poor O_2_ sensing. It was also linked to reduction in number of genes involved in stem development, number of proteins involved in cytoskeleton, lignin, and methionine metabolism along with two selenocysteine t-RNA synthases and a selenium (Se)-binding protein [106]. Copper shortage also impacted proteins involved in tetrapyrrole biosynthesis, isoprenoid biosynthesis, SAM biosynthesis and activation of the MOT1 gene facilitating Mo absorption [107]. Most Cu-deficient plants exhibit a faulty photosynthetic electron transport chain, as well as decreased non-photochemical quenching, which is consistent with altered plastocyanin function. Other Cu- related genes identified include SQUAMOSA promoter-binding protein-like 7 (SPL7), a Cu homeostasis transcription factor [108], Cu chaperone (antioxidant protein 1, ATX1) and one P1B-ATPase, involved in intracellular copper transport [106].

Cu toxicity significantly reduced the development of shoots and roots in wheat and enhanced lipid peroxidation [106]. Proteins involved in signal transduction, stress resistance, oxidative stress and production of energy were all found to be significantly increased, whereas proteins involved in carbohydrate metabolism, protein metabolism, activation of lipid oxidation and photosynthesis were all significantly reduced [109]. High amounts of Cu impeded the production and conversion of energy, according to proteomic investigations. The levels of SOD, CAT and nitrate reductase activities were all affected. In a proteomic study, heterotrophic Cu (II) stressed lipid accumulation in *Chlorella protothecoides* was found to be beneficial [110]. About 30 distinct proteins involved in glucose metabolism, carbon fixation, TCA cycle, lipid metabolism, protein biosynthesis, transportation and regulation, ATP and RNA biosynthesis, nucleotide metabolism, and ROS scavenging were also found in this study. Cu toxicity was analyzed in rice at the proteome level in a 2DE research on rice embryos, with 16 proteins connected to Cu stress, such as metallothionein-like protein, pathogenesis-related proteins, and GTP-binding protein Rab2 [111]. Copper toxicity, in collaboration with reactive oxygen species (ROS), altered more than 20 proteins involved in defense and stress response, as well as protein metabolism and modification, in rice seeds. Proteins such as disulfide isomerase (PDI), triose phosphate isomerase (TPI), and mannose 6-phosphate reductase (M6PR) were identified in varied Cu concentration treatments with or without H2O2 stress, revealing the capacity of rice seeds to respond to high levels of copper contamination [112]. 2DE analysis was used to determine the proteomic response of tolerant (B1139) and sensitive (B1195) varieties of rice roots to Cu.

Other studies have shown that ectopic expression of rice laccase genes (OsLACs), particularly OsLAC10in Arabidopsis, resulted in its tolerance to excess Cu and it was further verified in *E. coli.* Overexpression of OsLAC10 might assist plant endurance to Cu stress, presumably by preventing excessive Cu absorption through root lignification [113]. In grapevines under Cu toxicity stress, the expression of genes associated with ROS generation (RboH, AO) and superoxide scavenging (Fe-SOD, POD, CAT) systems were studied, revealing that desirable genes were up-regulated to overcome the oxidative stress generated by Cu [114]. The OsHMA4 gene, which is responsible for Cu accumulation in rice, was highly stimulated by high Cu accumulation in roots rather than shoots, suggesting that OsHMA4 plays a crucial role in Cu homoeostasis [74]. In *Arabidopsis thaliana*, Cu toxicity resulted in down-regulation of the SPL7 transcription factor (that plays a key role in plant growth and development) suggesting its putative role in Cu stress [115].

### 3.4. Iron

Fe insufficiency causes harm to agricultural output and quality, which indirectly impacts human health through the food chain, primarily vegetarians. To deal with Fe deficiency, plants have evolved complex systems to maintain cellular Fe homeostasis through reprogramming a variety of physiological, morphological, metabolic, and gene expressions [116]. To study Fe deficiency in plants, most proteomic studies utilized protein extracts from entire roots [65]. In *A. thaliana*, a proteomic investigation revealed the presence of coumarins that are critical for Fe acquisition [117]. The 2D-PAGE with IEF followed by (SDS)-PAGE is the most commonly used method to study the profile of protein changes during Fe deficiency [118]. Plants have evolved two techniques to deal with iron scarcity. In strategy I, the plant enhances Fe solubility by raising ATP-dependent acidification of the rhizosphere which helps in the reduction of Fe(III) to Fe(II), via increasing NADH availability along with inducing a Fe(II) transporter. The proteomic technique has primarily been applied to strategy I plants under iron deficiency conditions. Only three studies so far, to the best of our knowledge, have been reported on proteomic changes in strategy II plants: one in Fe-deficient rice shoots and roots, the second in root hairs of Fe-deficient maize (*Zea mays*), and the third in plasma membrane preparations of Fe-deficient maize roots [119]. In Strategy II synthesis and release of high-affinity iron chelators called phytosiderophores in the rhizosphere facilitate absorption of Fe (III) straight from the ground.

Although Fe is critical for plant growth, excess iron is highly harmful resulting in damage to plant tissue due to production of reactive oxygen species (ROS), which may attach to thiol, carboxyl, and imidazole groups in proteins, changing their structures and activities. Due to the antagonistic relationship between Fe and other vital metals, an excess of iron supply can lead to metal imbalance [120]. In certain circumstances, an excess of iron is used as a probe to discover the involvement of specific proteins involved in iron homeostasis [109,121].

Fe-induced genomic alterations in Lens culinaris were investigated and an up-regulation of Fe metabolism genes (Ferritin-1, BHLH-1 and IRT-1) were identified. Under Fe excess conditions, these genes maintain a balance between absorption and translocation of Fe [122]. In the leaves of the Japonica variety of rice, genes involved in hormonal control, senescence, absorption, transport, and stress oxidant were examined under Fe stress and were extensively characterized. The up- and down-regulation of genes (depending on the concentration used) were shown by microarray validation using RT-qPCR. Out of the 2525 associated genes, 1773 were found with known functions, of which 1720 were up-regulated, while 53 were down-regulated [123]. In Citrus and Arabidopsis, Fe deprivation causes differential gene expression by overregulating genes involved in cell wall manufacturing, ethylene and abscisic acid signal transmission, and Fe homeostasis [76,77]. These studies from different plant species provide a wealth of information about different genes and proteins that are crucial to combat the Fe-stress conditions.

### 3.5. Manganese

Manganese deficiency is more frequent in alkaline soils. A study was conducted in two barley varieties where PSII super complexes were shown to be drastically reduced in response to Mn shortage. Despite the breakdown of PSII, manganese insufficiency occurs with no obvious signs or symptoms [124]. Under Mn insufficiency, proteins that were involved in photosynthetic machinery remained unaltered, whereas increase in level of oxygen-evolving enhancer proteins in the soluble protein fraction was marked. These proteins became less linked with thylakoid membranes. Using a label-free, quantified proteomics method, researchers found that MnSOD1 level was significantly decreased and 270 proteins were changed in the soluble proteome of algae during Mn stress in Chlamydomonas [125].

Increased Mn^2+^ levels in soils can affect the absorption of other cations. Excess Mn^2+^ accumulates in the apoplast, where it is oxidized to Mn^3+^, which damages lipids and proteins by oxidizing them. The apoplast proteome of cowpea (*Vigna unguiculata*) was analyzed by 2D-PAGE and BN-PAGE under high Mn availability, revealing changes in pathogen-related proteins (PR proteins) and apoplastic peroxidases [2]. The formation of brown patches (Mn toxicity) is linked to an increase in peroxide. When compared to a Mn-tolerant cultivar of cowpea, apoplastic peroxidase activity was higher in the Mn-sensitive cultivar [126]. A study using proteomics approaches disclosed that *Camellia sinesis* can tolerate Mn toxicity in the presence of CsMTP8 metal transport protein, which is localized in plasma membranes of cultivar-LJCY [127]. A 2D-PAGE analysis of the Mn toxicity in the leaf proteome of poplar (*Populuscathayana*) revealed that about ten proteins involved in gene expression, and regulation, cell signaling, cell defense, cell death, heat shock, ROS scavenging, and photosynthesis were activated. In male poplar plant leaves, there was an increase in the production of four proteins (peroxiredoxin BAS1, trypsin/chymotrypsin inhibitor, HSP, and actin-1) and a decrease in expression of zinc finger protein. On the other hand, female poplar plants had higher levels of two proteins (2-cys thioredoxin BAS1, RuBisCo) and lower levels of one protein (actin-1).Proteomic studies on stylo (*Stylosanthesguianensis*) have shown that Mn detoxification, via malate synthesis and exudation, was mediated by SgMDH1 at both the transcriptional and protein levels [128].The genotype RY5 recorded the maximum ability to tolerate manganese toxicity and it was the genotype’s ability to regulate tens of proteins in roots and leaves involved in defense response (glutathione S transferase and antioxidant enzymes (catalase, superoxide dismutase, and peroxidase)) that gave the plant the ability to manage manganese [129]. Another proteomic research was conducted on the capacity of barley and rice to withstand manganese toxicity and it was found that older rice leaves had a stronger ability to bind manganese to the plant cell wall [130].

Mn deficiency-induced genes and their activities were discovered in Arabidopsis roots providing a part of a genome analysis. It also facilitated the detection of a large number of photosynthesis related genes in diverse pathways like Calvin cycle, glycolysis, TCA cycle, respiration, and fermentation [78]. Using the candidate genes AK251925.1, AK249774.1, and MLOC 77860.1, another study discovered that Mn regulates photosystem II subunits and that their co-localization is significant. In addition, two additional genes, AK357955 (protein phosphatase) and AK368229 (chlorophyll a/b binding protein), have been shown to be directly engaged in Mn-dependent regulation [131]. Two additional genes, AK357955 (protein phosphatase) and AK368229 (chlorophyll a/b binding protein), have been reported to be involved in Mn-dependent regulation [6]. *Citrus sinensis* and *Citrus grandis* were given high amounts of Mn for better understanding of Mn tolerance. The cDNA-AFLP analysis showed that expression of genes involved in numerous functions, such as ATPase production, leaf senescence, nucleic acid and protein metabolism, cell formation and transportation, were differentially expressed during the Mn stress conditions. Under Mn toxicity in *C. grandis*, genes involved in phosphorylation and dephosphorylation were both up- and down-regulated, indicating their balanced regulation towards the effects of oxidative stress [79].

### 3.6. Molybdenum

Deficiency of Mo in various forms affects the metabolism of plants at different levels. Mo deficit plants show uniqueness in phenotypic expression, like leaves with molting lesions, alteration in leaf morphology, whiptail formation or involuted lamellae [132].

Very few agricultural plants suffer from Mo toxicity. Plants like cauliflowers, tomatoes, legumes etc. subjected to the Mo toxicity were reported to have yellow to purple leaves. Mostly ruminant animals get affected by eating Mo toxic plants which causes Molybdenosis disease, eventually resulting in Cu deficiency. The solution to this is reducing the proportion of Mo uptake by plants from the soil by application of superphosphate or including sufficient proportions of Mo/Cu ratio in the animals’ diets.

Researchers have found that molybdenum shortage resulted in reduced expression of genes like Xanthine dehydrogenase (XDH), nitrate reductase (NR), aldehyde oxidase (AO), and sulfite oxidase (SO), in the leaves and roots of winter wheat. In wheat, Mo insufficiency increased the expression of TaCnx2 and TaCnx5, which could lead to an increase in molybdopterin (MPT) and cyclic pyranopterin monophosphate (cPMP) accumulation. It also reduced TaCnx1 expression, which is thought to diminish the production of molybdenum cofactor (Moco). The expression of TaAba3 in winter wheat was caused by a lack of Moco. Other genes involved in Mo deficiency include MOT1 and MOT2, with elevated expression in the shoot of plants, resulting in decreased Mo uptake and transport from the root to the shoot [133]. In the T-DNA insertional mutant of *A. thaliana*, mot1-1, Mo deficiency marginally increased the expression of CNX2, CNX5, CNX6, ABA3, and MOT2 [134]. In both leaves and roots, Mo shortage raised the levels of TaCnx5 mRNA. It also promoted TaAba3 expression in roots and leaves. Factors like light, nitrates and downstream N assimilation products, such as amino acids or C products, all stimulate NR mRNA production. In the leaves of winter wheat, Mo deprivation increased NR gene expression but NR activity deteriorated. Drop in the NR activities could be linked to a lack of Moco or be a compensatory response to the decrease in nitrate absorption under Mo deficiency conditions [135]. It has also been studied that Mo shortage resulted in decreased XDH activity, as well as its gene expression, implying that Mo deficiency limits purine catabolism and promotes senescence. In brief, via controlling Mo enzyme activity and gene expression, Mo shortage may suppress purine catabolism, sulfite detoxification, nitrate assimilation, and ABA biosynthesis.

### 3.7. Zinc

2D gels or label-free quantitative proteomics were used to investigate zinc insufficiency. Transcriptomic techniques have been used to investigate Zn insufficiency, with results indicating that Zn shortage strongly regulates transcription factors. Zn finger proteins account for just 0.7 percent of total protein content and are included in a vast number of transcription factors [107]. Under zinc deprivation, there was an increase in two defensin-like (DEFL) family proteins in Arabidopsis roots [136]. In the single-cell green algae Chlamydomonas, two COG0523 domain-containing proteins were accumulated in significant amounts, but zinc-containing carbonic anhydrase (CAH) isoform 1 was decreased [125]. CAH1 null mutants in Chlamydomonas did not show severe growth abnormalities in low-CO_2_ growth circumstances, indicating that CAH1 is a dispensable enzyme for a cell in desperate need of internal Zn for recycling and redistribution.

Zinc toxicity causes continual inhibition of protein disulfide isomerase and triosephosphate isomerase [136]. Proteomic studies on the root proteome of *Populuseur- americana* reported changes as a result of Zn toxicity, and it was discovered that there was a rise in antioxidant proteins system, a reduction in carbohydrate/energy metabolism proteins, and a shift in amino acid metabolism [136]. Excessive zinc also inhibited anthocyanidin synthase, a crucial enzyme in the phenylpropanoid pathway, affecting poplar polar root secondary metabolism [136]. An analysis of Zn’s toxic effects in the mitochondrial proteome of sugar beet roots revealed a close link between Zn toxicity and ROS generation [137]. Zn accumulates in *Arabidopsis halleri* leaves, and when the plant was exposed to metals or metals and microbes, photosynthesis-related proteins were up-regulated. This demonstrated that metal accumulation in shoots demands energy [138]. The interaction between excess Zn and Fe shortage were also studied in Arabidopsis roots using iTRAQ for microsomal proteins because of the resemblance in symptoms of Zn toxic plants with the symptoms of iron deficient plants [136,139]. From the studies, it was identified that proteins, especially phytochelatins and nicotianamine helped in chelation of excess zinc in the cytosol and during transport. In *A. thaliana*, the overexpression of the NcZNT1 gene resulted in increased tolerance against excessive Zn exposure. Under Zn shortage, the ZNT1 Zn transporter, AtZIP4, was substantially up-regulated in the pericycle, cortex and endodermis [140] and could be a crucial partner in maintaining homeostasis under deficit stress conditions.

## 4. Conclusions

Micronutrients play a vital role in plant growth and development. The control of processes like respiration, photosynthesis and other systems may be affected by an excess or shortage of micronutrients. The metabolism of various proteins and the biological activity of the soil are both slowed down by micronutrient shortage. However, when available in excess concentration micronutrients can be highly detrimental to plant health. Due to excessive use of fertilizers and contamination of soils with heavy metals, the percentage of cultivable land is decreasing across the world. With the advancement of genomic and proteomic technologies, it is highly feasible to understand the underlying mechanisms of nutrient stress responses in various crops. Research around the world has studied the effect of different micronutrient toxicities and has provided a wealth of information that could be used to improve the stress tolerance of commercial crops. In most of the studies it has been observed that genes involved in photosynthesis, respiration, and detoxification systems are crucial. The control of these processes and other systems may be affected by an excess or shortage of certain micronutrients. Whole-cell extracts of soluble proteins, sub-proteomes, and membrane proteins have so far been described using gel-based and gel-free techniques. Plasma membranes and microdomains, which are essential targets for nutrition transport, stress signaling, and adaptability, should receive more attention in future studies.

## Figures and Tables

**Figure 1 plants-11-02424-f001:**
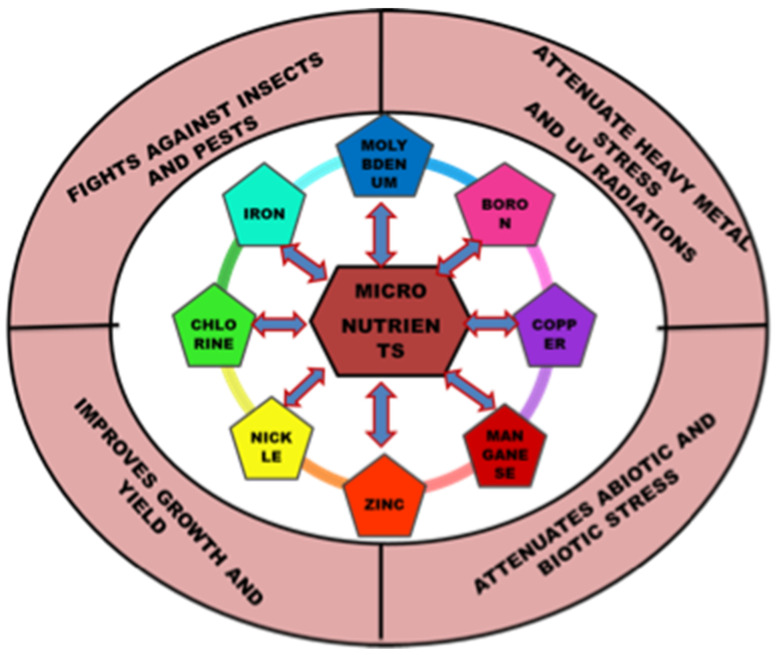
Versatile role of micronutrients in plant’s growth and development.

**Figure 2 plants-11-02424-f002:**
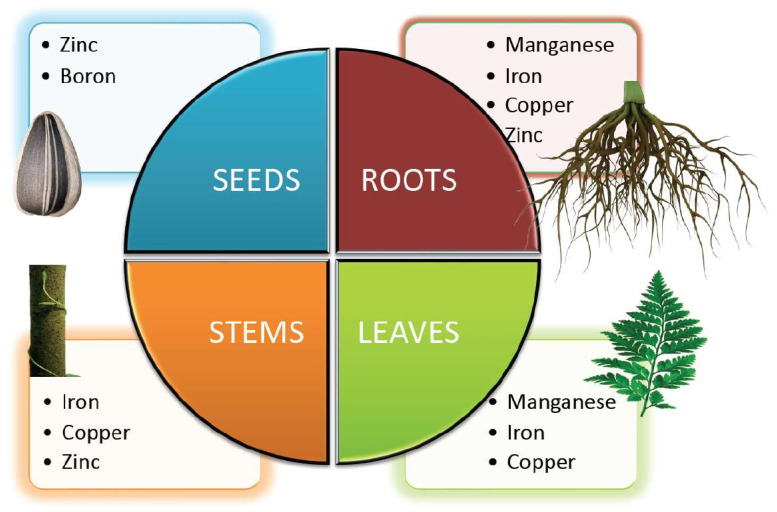
Micronutrients involved in the growth and development of different parts of the plant.

**Figure 3 plants-11-02424-f003:**
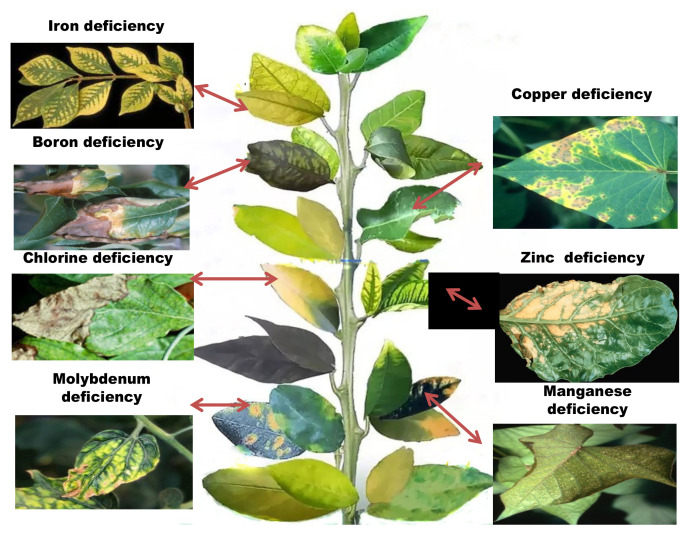
Symptoms of micronutrient deficiencies in plants.

**Figure 4 plants-11-02424-f004:**
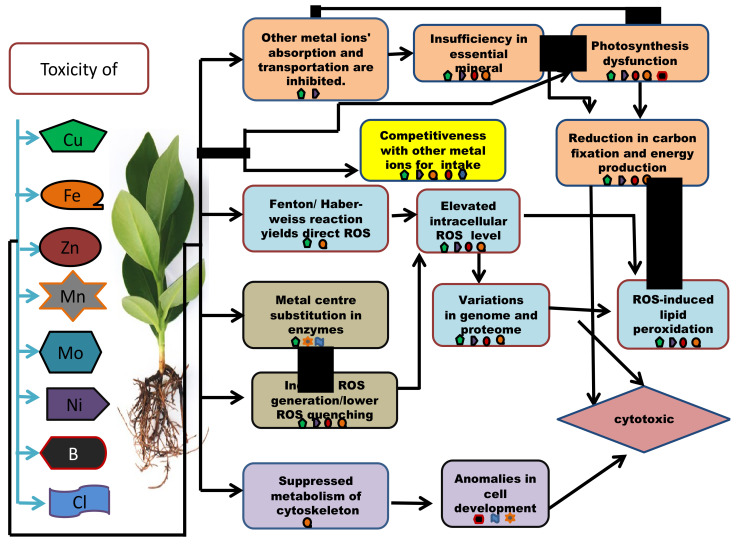
An outline presenting the molecular and the biochemical mechanisms involved in micronutrient cytotoxicity in plants.

**Table 1 plants-11-02424-t001:** Micronutrients, their modes of intake and their concentrations in the leaves.

S. No.	Micronutrient	Year of Discovery	Ionic Form of Intake	Normal Value (ppm)	Deficient Value (ppm)	Toxic Value (ppm)
1	Boron (B)	Warington (1937)	H_2_BO_3,_ H_2_BO_3_^−^, HBO_3_^2−^, BO_3_^3−^	10–20	5–10	50–200
2	Chlorine (Cl)	Broyer et al. (1954)	Cl−	100–500	Less than 100	500–1000
3	Copper (Cu)	Lipman and Mackinney (Sachs 1931)	Cu^2+^	5–30	2–5	100–200
4	Iron (Fe)	Sachs (1860)	Fe^2+^, Fe^3+^	100–500	Less than 50	More than 500
5	Manganese (Mn)	Mchargue (1922)	Mn^2+^	20–300	15–20	300–500
6	Molybdenum (Mo)	Arnon and Stout (1939)	MoO_4_^2−^	0.1–2.0	0.03–0.15	More than 100
7	Zinc (Zn)	Sommer and Lipman (1926)	Zn^2+^	27–150	10–20	100–400

**Table 2 plants-11-02424-t002:** Summary of micronutrient functions and deficiency symptoms.

S.No.	Micronutrients	Representative Constituents/Proteins	Functions	Symptoms of Deficiency	Probable Cause of Deficiency & Method of Correction	References
1	Zinc (Zn)	Cu–Zn superoxide, Peptide deformylase, enzyme carbonic anhydrase (CA), α-Mannosidase, Matrix metalloproteinase alcoholic dehydrogenase, and superoxide dismutase (SOD)	role in nitrogen metabolism and photosynthesis, controls the concentration of auxin in plants, increases seed viability and seedling vigor, protection against abiotic and biotic stresses	Reduced vigor, chlorotic leaves, white streaks parallel to leaf blade, slow growth, restricted RNA and protein synthesis	Low Zn in soil, high soil pH—lower soil pH, apply foliar spray or add Zn to soil.	[8,9,10,11,12,13,14,15,16]
2	Copper (Cu)	Plastocyanin Cu-Znsuperoxide, Ascorbate, Cu-metallothionein, biosynthesis oxidase, Mo-cofactor, dismutase, polyphenol or catechol oxidase, tyrosinase, laccase, Cytochrome-C oxidase, Ethylene receptor ascorbic oxidase and Polyphenol oxidase	saves plants from diseases, improves the fertility of male flower, concerned with the oxidation of iron in plants	Leaf tips dries, break down and dies, ragged leaves, reduced growth	Low soil Cu, high organic matter—apply foliar spray or add Cu to soil	[17,18,19,20,21,22,23,24,25,26]
3	Iron (Fe)	Aconitase, dismutase, Xanthine dehydrogenase, Ferredoxin, porphyrin NADH, Succinate, Leg hemoglobin, heme and heme enzymes peroxidase oxidase, dehydrogenase, Nitrate reductase oxidoreductases, Thioredoxin reductase, Cytochrome P450, Aldehyde oxidase, Catalase, Nitrite reductase, Lipoxygenase, Alternative oxidase, Fe-superoxide Ferritin and other functional metallic proteins	Present in haemoglobin of the leguminous root nodules, leg-haemoglobin and is involved in nitrogen fixation as a constituent of ferredoxin	Interveinal, creamy chlorosis on apical leaves, stunted shoots, reduced yield	Waterlogged soil, over fertilized, excess of elements like Mn—spraying plant with iron rich fertilizer, chelated iron powder or blood meal directly to the soil	[27,28,29,30,31,32,33,34,35,36,37,38,39,40,41,42,43,44,45,46]
4	Manganese (Mn)	Malic enzyme, Mn-superoxide, amidohydrolase, primary component of water-splitting enzyme related to photosystem II, PEP carboxylase, Allantoate, Isocitratelyase, dismutase PEP-carboxykinase	Involved in tricarboxylic acid cycle in oxidation and reduction reactions, activates several enzymes such as oxidoreductases, hydrolases and lyase, also autocatalyzes isocitrate dehydrogenase, malic dehydrogenase, glycocyaminase and D-alanyl synthase.	Reduced quality and yield, stunted plants, intervenial chlorosis in leaves, yellow cast in deficient areas. Death of basal leaves, decreased cold hardiness, growth of lateral roots stopped, inhibition of nitrate metabolism.	Low soil Mn, high soil pH due to over liming—lower soil pH, apply foliar spray or add Mn to soil	[47,48,49,50,51,52]
5	Boron (B)	Rhamnogalacturonan II	increases cell wall thickness and flower production, as well as retention, pollen tube elongation and germination, along with seed and fruit development. It also helps in the translocation of photosynthates. It inhibits IAA oxidation and gives drought tolerance to crops	Thick and leathery old leaves, shoot tip death, rosette leaves with short internodes, excess branching, short, twisted and/or ruptured petioles, vegetables with hollow heart, small and deformed/no fruits with cork spots chlorosis, stubby roots, inhibited nitrogen metabolism.	Low soil B especially on sandy soils or light textured soils—apply foliar spray or add B to soil	[53,54,55,56]
6	Molybdenum (Mo)	nitrogenase, sulphite oxidase, nitrate reductase Aldehyde oxidase and xanthine oxidase/dehydrogenase	aids in the synthesis of ascorbic acid, formation of pollens and anthers, acts as a remedy to excessive copper, manganese and zinc	stunted growth with twisted stems, leaves turning pale green, necrotic area in leaves along the mid rib between veins and along leaf edges	Low soil pH, low Mo content in soil—inoculate seed with Mo, apply foliar spray or add Mo to soil	[33,34,57,58]
7	Chlorine (Cl)	Oxygen-evolving complex Seismonastic movement	activates enzymes that are involved in starch utilization which affects germination and energy transfer. Inmoisture-stress conditions chlorine helps in the movement of water into cells and maintenance of that water. Chlorine also controls the opening and closing of stomata on leaf surfaces	stunted/restricted growth, stubby roots, interveinal chlorosis, non-succulent tissue, wilting	Low soil Cl especially in soils subjected to leaching—apply Cl containing fertilizer	[55,59]

**Table 3 plants-11-02424-t003:** Micronutrient toxicity.

S. No.	Micronutrients	Type	Symptoms of Excess Usage	References
1	Zinc (Zn)	Metal	Inhibited root growth, young leaf chlorosis, interveinal mild chlorosis in young leaves starting from base and spread towards apex followed by reddish brown coloration, rolling of leaves	[61]
2	Copper (Cu)	Metal	Reduced vigor, inhibited root growth/root damage, older leaves develop orange or pink coloration followed by severe rolling of leaf margins due to loss in turgidity	
3	Iron (Fe)	Metal	Reduced yield, bronzing and stippling of leaves, and, in some plants, acid is secreted from the roots	
4	Manganese (Mn)	Metal	Tissue injury, leaf sheath and lower parts of stem in cereal normally consist of minute brown spots, legumes develop brown or purple spots over leaf margin, deficiency symptoms of other nutrients	
5	Boron (B)	Metalloid	Toxicity results in dark brown speckles or necrosis on the edge of older leaves, cupped and wrinkled young leaves	
6	Molybdenum (Mo)	Metal	Leaf malformation, tints of golden yellow or blue color in leaves	
7	Chlorine (Cl)	Non metal	Death of leaf margin, leaves are reduced in size and number, have bronze or yellow coloration with brown or scorched leaf margins.	

**Table 4 plants-11-02424-t004:** Proteomic response of plant to micronutrient stress.

Metal	Plant Species	Plant Part	Extraction Method	Protein Name	Function	Regulation	References
Boron (B)	*Citrus grandis*	Root	iTRAQ	Alcohol dehydrogenase 1	Energy metabolism	Down	[62]
				Serine/arginine- rich 22	Nucleic acid metabolic process	Down	
				Clathrin light chain protein	Cellular cytoskeleton and transport	Up	
				Peroxiredoxin IIF	Cellular response to stress	Down	
				Phospholipase C2	Signal transduction	Down	
	*Arabidopsis thaliana*	Leaf	2-DE, LC-MS/MS	Rubisco activase	Photosynthesis	Up	[63]
				Actin 7	Metabolism	Down	
				Fructose-bisphosphatealdolase	Energy metabolism	Up	
				Glycolate oxidase	Photosynthesis	Down	
Coppe (Cu)	*Sorghum bicolor*	Leaf	2-DE, MALDI-TOF MS	Thymidine kinase	Protein translation and synthesis	Down	[64]
				Thaumatin-like protein	Stress and defense	Up	
				Maturase K	Growth and development	Down	
				Alcohol dehydrogenase	Oxidation– reduction process	Up	
	*Oryza sativa*	Root	2-DE, MALDI-TOF MS	Putativeperoxidase	Antioxidation and detoxification	Up	[65]
				Putative cold shock protein-1	Transcriptionalregulation	Up	
				Putative elongationfactor EF-2	Protein synthesis	Down	
				Glutamine synthetase shoot isozyme	Amino acid synthesis	Down	
	*Allium cepa*	Root	2-DE, MALDI-TOF MS	Glutaredoxin	Defense	Up	[66]
				Ran-binding protein 1	Protein synthesis	Down	
				Cinnamoyl-Co-A-reductase 1	Cell wall synthesis	Down	
				Proliferation-associated 2 g4	Cell cycle and DNA replication	Up	
Iron (Fe)	*Arabidopsis thaliana*	Root	Itraq, LC-MS	Oxidoreductase	Hormone metabolism	Up	[67]
				WAKL4, WAK- like receptor- like kinase	Signaling	Up	
				FRO3, ferric chelate reductase 3	Metal handling	Up	
				SAPX, Stromal ascorbate peroxidase	Redox	Down	
				IRT3, Iron regulated transporter3	transport	Down	
	*Zea may*	Root	LC-MS/MS	Aquaporin PIP2-2	Transport proteins	Down	[68]
				Gibberellin receptor GID1L2	Signaling proteins	Down	
				Aldolase	Metabolism	Up	
				Actin-2	Cytoskeleton	Down	
				Callreticulin2	Protein folding	Up	
	*Cucumissativus*	Root	2-DE, ESI-LC-MS	Phosphoglycerate kinase	Glycolysis	Up	[69]
				Malate dehydrogenase	Carbohydrate-related metabolism	Up	
				Alanine aminotransferase	Nitrogen-related metabolism	Up	
				Xylan 1,4-beta-xylosidase	Metabolism of sucrose	Down	
Manganese (Mn)	*Citrus grandis*	Root	2-DE, MALDI-TOF MS, LTQ-ESI-MS/MS	Acetohydroxyacidisomeroreductase	Protein metabolism	Down	[70]
				Maturase K	Nucleic acid metabolism	Up	
				Alcohol dehydrogenase	Carbohydrate metabolism	Up	
				Iron- sUperoxide dismutase	Stress response	Down	
				Valosin- containing protein	Cell transport	Down	
Zinc (Zn)	*Lactuca sativa*	Leaf	LC-MS/MS, MALDI-TOF	MPBQ/MSBQ transferase	Photosynthesis	Up	[71]
				Cytosolic fructose-1,6-bisphosphate	Energy metabolism	Up	
				Putative cellulose synthase	Cell wall metabolism	Down	
				Phenylalanine ammonia-lyase	Phenylpropanoids biosynthesis	Down	

**Table 5 plants-11-02424-t005:** Examples of genomic responses of plants to nutritional stress.

Metals	Plant Species	Tissue	Method of Identification	Gene	Functions	Regulation	References
Boron (B)	*Arabidopsis thaliana*	Root	qRT-PCR	SHB1/HY1	B-tolerance	Up	[72]
	*Oryza sativa*	Root	Microarray, qRT-PCR	LOC_Os08g30740	ABC transporter	Down	[73]
				LOC_Os10g30156	Starch synthase	Down	
				LOC_Os10g30080	Xylosyltransferase	Down	
	*Citrus sinensis*	Leaf	Illuminasequencing, qRT-PCR	Ciclev10017846m	Ubiquitin-protein ligase activity	Down	[74]
				Ciclev 10012377m	Transcription factor	Up	
				Ciclev 10009779 m	Blue copper protein	Up	
Zinc (Zn)	*Arabidopsis thaliana*	Whole plant	qRT-PCR	PDR8	Phytochelatin synthesis	Up	[75]
				ABCCI	Metabolism	Down	
Iron (Fe)	*Citrus sinensis*	Whole plant	qRT-PCR	Cs3g19420	Ethylene-responsive transcription factor	Up	[76]
				Cs4g12540	Ethylene synthesis	Down	
				Cs9g02930	Flavone synthase	Down	
	*Arabidopsis thaliana*	Root	qRT-PCR	FRO2	Iron homeostasis	Up	[77]
Manganese (Mn)	*Arabidopsis thaliana*	Root	qRT-PCR and RNA sequencing	AT5G05960	Bifunctional inhibitor	Down	[78]
				AT4G39940	Glucosinolate	Up	
				AT1G15820	metabolism Photosynthesis	Up	
	*Citrus sinensis*	Leaf	cDNA-AFLP and qRT-PCR	TDF #065-1	Energy metabolism	Up	[79]
				TDF #073-1	Cell transport	Down	
				TDF #103-2	Stress responses	Down	
Copper (Cu)	*Arabidopsis thaliana*	Whole plant	qRT-PCR	OsLAC10	Laccase activity	Up	[80]
				AtLAC11	Lignin biosynthesis	Up	
	*Oryza sativa*	Whole plant	Western blotting, qRT-PCR	OsHMA4	Copper accumulation	Down	[74]

## Data Availability

Not applicable.

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
