# Peer review of "Proteomic and Genomic Studies of Micronutrient Deficiency and Toxicity in Plants"

_plants, 2022, doi:10.3390/plants11182424_

Round 1
Reviewer 1 Report
The review article can be accepted after considering the following items:
each elements in Micronutrients toxicity (table 3 ) should be cited. In addition, each constituents of the micronutrients functions and deficiency symptoms (table 2) should be also cited.
The review article should be enhanced with more figures.
Finally, Table should be fabricated to highlight the purpose of creating such review when compared with other published articles.
Author Response
We sincerely thank all the reviewers for taking time to review our manuscript and giving their valuable suggestions and comments for the improvement of our manuscript.
Author’s response:
Reviewer 1-
Comment 1: Each element in Micronutrients toxicity (table 3) should be cited.
Response 1: Thank you for your valuable suggestion. We have revised the table and added the references for each micronutrient.
Comment 2: Each constituent of the micronutrients functions and deficiency symptoms (table 2) should be also cited.
Response 2: Thank you for your valuable comment. We have revised it as per your suggestions.
Comment 3: The review article should be enhanced with more figures.
Response 3: Thank you for your valuable comment. We have added two more figures
Comment 4: Table should be fabricated to highlight the purpose of creating such review when compared with other published articles.
Response 4: Thank you for your valuable comment. We tried out best to compile all the relevant research performed in the field of plant micronutrient work in the form of these tables. We hope the compilation is satisfactory.
Reviewer 2 Report
Dear Authors,
Dear Editor,
To me, the manuscript is full of errors, weak wording and contradictory statements. Even chemical symbols (ions) contain typographic errors. Thus, I was not able to go far beyond the first pages with my remarks:
p1 l33 first "and" may be replaced by a comma
p1 l33 "better" refers to what?, may be deleted
p1 l35 "is"
p1 l38 "total" is not necessary
p1 l40 "of each micronutrient crop" - what does it mean?
p1 l44 "total micronutrients available" - normally, we distinguish between "total" and "available" (a fraction is available)
p2 l54 "Plants seem to be unable to detect a deficit because micronutrient availability is impacted by.." is contradictory
to "When nutrient demand exceeds the rate of supply, the plant frequently switches to alternative metabolic pathways of l 51 before
Most probably, the task was far too ambitious. Even for one selected element it would be a very big challenge to prepare a review paper in the proposed context.
Sorry for that.
Author Response
We sincerely thank all the reviewers for taking time to review our manuscript and giving their valuable suggestions and comments for the improvement of our manuscript.
Author’s response:
Reviewer 2-
Author’s highly regret to disappoint the reviewer. We tried our best to address the comments and suggestions made by the reviewer. However, as the comments were not specific, we could not address the actual concerns of the reviewer.
Comment 1: Even chemical symbols (ions) contain typographic errors.
Response 2: Thank you for your valuable comment. In the manuscript we have mentioned only the elemental form of micronutrients. The ionic form that is taken up by the plant is mentioned in the table 1.
Comment 2: p1 l33 first "and" may be replaced by a comma
Response 2: Thank you for your valuable comment. We have changed the sentence as per your suggestions.
Comment 3: p1 l33 "better" refers to what?, may be deleted
Response 3: Thank you for your valuable comment. In the sentence “better” means “easy access to the micronutrients”. We have modified the sentence for the better understanding of the readers.
Comment 4: p1 l35 "is"
Response 4: Thank you for your valuable comment. We have made necessary corrections.
Comment 5: p1 l38 "total" is not necessary
Response 5: Thank you for your valuable comment. We have made necessary changes.
Comment 6: p1 l40 "of each micronutrient crop" - what does it mean?
Response 6: Thank you for your valuable comment. We have added “of each micronutrient in the crop” to the sentence which was missing.
Comment 7: p1 l44 "total micronutrients available" - normally, we distinguish between "total" and "available" (a fraction is available)
Response 7: Thank you for your valuable comment. We have made the necessary corrections.
Comment 8: p2 l54 "Plants seem to be unable to detect a deficit because micronutrient availability is impacted by.." is contradictory
to "When nutrient demand exceeds the rate of supply, the plant frequently switches to alternative metabolic pathways of l 51 before
Response 8: Thank you for your valuable comment. We have made the necessary changes.
Reviewer 3 Report
This manuscript was well-written, all figure and table in this manuscript were arranged in high quality. I agrre to accept it.
Author Response
We sincerely thank all the reviewers for taking time to review our manuscript and giving their valuable suggestions and comments for the improvement of our manuscript.
Reviewer 3-
Comment 1: This manuscript was well-written, all figure and table in this manuscript were arranged in high quality. I agree to accept it.
Response 1: Thank you very much for reviewing our manuscript and accepting it for publication purpose.
Round 2
Reviewer 2 Report
Dear All,
Unfortunatel, I can not see any substantial improvement,
Best regards
Refereee
Author Response
Dear Reviewer,
Greetings! We have made the necessary changes requested by all the reviewers and the editor. We kindly request you to consider our revised manuscript for the review.
Thanks
Sowmya
